# Tow Deformation Behaviors in Resin-Impregnated Glass Fibers under Different Flow Rates

Sung-Woong Choi [1], Sung-Ha Kim [2], Mei-Xian Li [3], Jeong-Hyeon Yang [1] and Hyeong-Min Yoo [4],*

[1] Department of Mechanical System Engineering, Gyeongsang National University, Tongyeonghaean-ro, Tongyeong-si 53064, Korea; younhulje@gmail.com (S.-W.C.); jh.yagi@gnu.ac.kr (J.-H.Y.)
[2] Materials Technology Department, Ebara Corporation, Fluid Machinery & Systems Company 4-2-1 Honfujisawa, Fujisawa-shi, Kanagawa 251-8502, Japan; kim.sungha@ebara.com
[3] School of Textile and Clothing, Nantong University, Nantong 226019, China; lmx321@ntu.edu.cn
[4] School of Mechanical Engineering, Korea University of Technology and Education (KOREATECH), 1600, Chungjeol-ro, Byeongcheon-myeon, Dongnam-gu, Cheonan 31253, Korea
* Correspondence: yhm2010@koreatech.ac.kr

**Abstract:** With the rapid development of high-performance fibers such as carbon, enhanced glass fibers in structural applications, the use of fiber-reinforced composite (FRC) materials has also increased in many areas. Liquid composite molding (LCM) is a widely used manufacturing process in composite manufacturing; however, the rapid impregnation of resin in the reinforcing fibers during processing poses a significant issue. The optimization of resin impregnation is related to tow deformations in the reinforcing fibers. The present study therefore focuses on this tow deformation. The permeability behaviors in double-scale porous media were observed under different flow rates and viscosity conditions to examine the overall tendencies of structural changes in the reinforcement. The permeability results showed hysteresis with increasing and decreasing flow rate conditions of 50–800 mm$^3$/s, indicating structural changes in the reinforcement. The tow behaviors of the double-scale porous media with respect to the thickness and flow rate were investigated in terms of the representative indices of the minor axis (tow thickness) and major axis. The minor axis and major axis of the tow showed decreasing and increasing trends of 2–5% and 2%, respectively, with minimum and maximum values at different positions along the reinforcement, affected by the different hydrodynamic entry lengths. Finally, the deformed tow behavior was observed microscopically to examine the behavior of the tow at different flow rates.

**Keywords:** liquid composite molding (LCM); tow deformation; permeability; hysteresis

## 1. Introduction

With the rapid development and increased use of high-performance fibers such as carbon, enhanced glass fibers in critical structural applications, high-strength and lightweight fibers are in high demand. With these trends, the use of fiber-reinforced composite (FRC) materials has increased in many areas such as structural applications, including rehabilitation and retrofitting, mainly because of their superior properties of high strength, corrosion resistance, and low weight [1,2]. Many researchers [3–6] have attempted to use FRC manufacturing process to fabricate especially for the complex designs. The materials for FRCs was consisted of use a resin, which can either thermosetting or thermoplastic, and a fibrous reinforcement, typified by glass fibers and carbon fibers. In fabricating a composite material, the mixing, interface, and interactions between the components must be considered.

To replace classical materials with composite materials, the mechanical properties of the composite must satisfy industrial standards and the fabrication process must be optimized. Various composite manufacturing processes have been developed for these purposes. One proposed manufacturing method for FRCs is liquid composite molding

(LCM), where a liquid-type resin is injected into a mold. Resin transfer molding (RTM) is a popular LCM process, in which fibers as continuous reinforcements are installed in the mold before the injection. The process is accomplished after curing the resin with a cross-linking reaction. The general merits of RTM are low cost and low energy consumption in forming products with complex geometries. The mold and tools are simple relative to those in other molding processes because of the relatively low applied pressure and temperature conditions. Because RTM uses a mold with the reinforcing fibers inside, the product design can be more complex compared to those achievable in other processes, such as compression molding.

The issues of LCM include its long production cycle such as autoclave processing and impregnation problems during the process. The long manufacturing times and cycles are mainly related to the resin curing time. Methods for decreasing the manufacturing time have been introduced that use rapid-curing resins. For the problem of resin impregnation in the fiber reinforcement, the flow behavior of the resin during injection into the reinforcements must be investigated.

The identification of impregnation problems through analyzing the flow behaviors during the LCM process has been attempted with a wide range of research techniques from macroscopic predictions of the mold flow to numerical predictions utilizing a porous media flow. Kang et al. [7] suggested a modified control volume finite element method to yield a smoother flow front and reduce the pressure error at the flow front using the conventional fixed-grid method. Masoodi et al. [8] conducted a numerical simulation of liquid composite materials using natural fiber preforms. Pillai et al. [9] suggested more accurate numerical methods to solve the governing equations and developed a multiscale model for unsaturated flow in LCM. Han et al. [10] proposed a model to simulate flow-induced fiber mat deformation for the impregnation problems.

Moreover, the deformation problem of reinforcement was related with hydrodynamic condition of injected resin during LCM processing. Regarding hydrodynamic perspective of injected resin, lots of studies was examined to deal with deformation problem of reinforcement. Hautefeuille et al. [11] experimentally examined the full-field magnitude of the flow-induced in-plane deformation of saturated fibrous reinforcement occurring during the compression. Bodaghi et al. [12] proposed a model for fiber-tow washout anticipation during HP-RTM, comparing the fluid and the friction forces. Endruweit et al. [13] experimentally investigated the effects of hydrodynamically induced textile deformation during resin injection in LCM processes for various fiber volume fractions in a rectangular flow channel with linear injection gate. Parnas et al. [14] proposed model of hydrodynamically induced preform deformation to assess the conditions where preform deformation is expected to play a dominant role during mold filling including preform material properties such as stiffness, permeability, and clamping pressure, as well as the process injection pressure and the mold geometry. Francucci et al. [15] has attempt to characterize the compaction behavior of jute woven fabric preforms in liquid composite molding processes. Michaud [16] presented several study cases for the non-saturated resin flow in liquid composite molding processes.

Especially for the tow deformation phenomenon, several investigations have been conducted. The tow-deformation mechanism was investigated by Hristov and Vasileva [17], who examined the mechanical properties and deformation mechanisms of polypropylene (PP)/wood fiber (WFb) composites modified with maleated PP as a compatibilizer and styrene-butadiene rubber as an impact modifier. The deformation mechanism in the unmodified PP/WFb composite was characterized by fiber pullout, debonding, and cavitation of the matrix, which resulted in brittle fracture. Xue et al. [18] developed an integrated micro/macro-constitutive model to predict the behavior of woven composites during large deformations based on the geometric parameters of fibers, yarns, and unit cells; the material constants of composite constituents; and the yarn orientation. Pazmino et al. [19] researched the formability of a single layer E-glass non-crimp 3D orthogonal woven reinforcement. Huang et al. [20] studied the mesoscopic deformations



of 2D glass woven fabrics by X-ray microtomography under transverse compaction for a range of fiber volume fractions encountered in high performance composite applications. Skordos et al. [21] developed a simplified finite element model validated for the forming/draping of pre-impregnated woven composites, incorporating the effects of wrinkling and strain rate dependence.

Although many investigations have been conducted to evaluate tow deformation phenomenon [17–21], a limited number of studies focused mainly on analytic and numerical model for the deformation of reinforcement of composite [18,21]. Few have focused on the tow deformation mechanism in the LCM process experimentally with visual approach [20]. The objective of our study was to observe tow bundle behavior experimentally and with visual approach since the reasonable explanation of tow deformation behavior should be based on the understanding of tow bundle itself.

The present study focused on the tow deformation phenomenon in the reinforcing fibers. The flow behavior in double-scale porous media was experimentally investigated under different hydrodynamic force effects, flow rates, and viscosities. The permeability behaviors in the double-scale porous media were examined with the hydrodynamic force effect to observe tow deformation. The tow behavior was analyzed under different flow rate conditions considering two representative indices: tow thickness (minor axis) and the major axis. Finally, a microscopic analysis was conducted to observe the tow behavior in the reinforcement.

## 2. Background and Theory

LCM processes are widely used to manufacture composite materials for automobile and ship applications in a cost-effective manner. RTM and vacuum-assisted RTM are representative LCM manufacturing processes. One of the main technologies in LCM is the impregnation of fiber reinforcements with a liquid resin. Resin flow in the LCM process can be regarded as a Newtonian flow in a porous medium, which can be modeled using Darcy's law:

$$u_i = -\frac{K_{ij}}{\mu} \frac{\partial p}{\partial x_j}, \tag{1}$$

where $u_i$, $K$, $\mu$, and $p$ represent the volume-averaged resin velocity, permeability tensor of the reinforcement, resin viscosity, and fluid(resin) pressure, respectively. The permeability is a key parameter for determining resin flow in the porous media of reinforcement. For determining the permeability, measurements of the saturated and unsaturated permeability are used. The former measures the pressure difference between two distinct points at a given flow rate under steady-state flow conditions in a fully saturated reinforcement in uni-directional flow [22]:

$$K_{sat} = \frac{\mu \frac{Q}{A}}{\left(-\frac{dP}{dx}\right)} = \mu \frac{Q}{A} \frac{L}{(P_1 - P_2)}, \tag{2}$$

where $K_{sat}$, $L$, $P_1$, and $P_2$ are the saturated permeability, the distance between the two pressure sensors, and the fluid pressure values at the distinct points 1 and 2, respectively. The unsaturated permeability in uni-directional flow is determined by following the flow front over time as the reinforcement is impregnated [22]:

$$K_{unsat} = \frac{\left(1 - V_f\right)\mu}{2P_1 t} \{L(t)\}^2, \tag{3}$$

where $K_{unsat}$, $V_f$, $t$, and $L(t)$ are the unsaturated permeability, fiber volume fraction, time instant, and distance between the liquid inlet and the flow front as a function of time, respectively.



It should be noted that for both of these permeability measurement methods, the pressure gradient is obtained assuming a linear pressure distribution, as expressed in the following relationship:

$$\frac{dP}{dx} = -\frac{(P_2 - P_1)}{L},$$ (4)

Generally, the permeability value is assumed to have the same value for a given fiber reinforcement type, regardless of the measurement method. However, it has been reported that the saturated and unsaturated permeability values can differ for the same reinforcement [23–27]. In many cases, the saturated permeability values exceed the unsaturated permeability values [23–28]. Many possible reasons for this discrepancy have been reported; capillary effects, void entrapment, and fiber deformation problems, etc. Among them, present study focused on the fiber deformation during the mold filling process as one of the contributors to the discrepancy. Therefore, the same value of saturated and unsaturated permeability is related to the assumption that fiber tows was not deformed during the resin impregnation.

## 3. Experimental

To observe the flow behavior in double-scale porous media, a non-crimped unidirectional glass fiber (Owens Corning, Ohio, USA) was used; its properties are shown in Table 1. In the reinforcement, each tow consisted of 1000 filaments with diameters of 16.5 μm. The horizontal and vertical dimensions of the tow cross-section were 1.5 and 0.5 mm, respectively. Silicone oil (KF-96, Shin-Etsu, Japan) was used as the injection working fluid, and its physical properties are listed in Table 2. Experiment was conducted with a rectangular mold measuring 500 mm (W) × 55 mm (L) × 3 mm (T), where W, L, and T are the width, length, and thickness of mold, respectively. The upper plate of the mold was 25 mm thick tempered glass, allowing observation of the fiber tow behavior during fluid flow. In the bottom part of the mold, pressure transducers (Sensys, Korea) with a measurement range of 0–0.1 MPa and a sensor accuracy of 0.030% were installed along the flow direction at 120 mm intervals from the inlet. The fluid was injected at a constant flow rate into the inlet in the mold using a fluid injection pump. The pressure data from the pressure transducers were recorded using a data logger (Keithley 2700, Ohio, US); measurements were repeated thrice for each condition. A schematic of the experimental apparatus for the observation of tow behavior in double-scale porous media is shown in Figure 1. The tow behavior was observed using an optical microscope (Olympus Optical Co., Ltd., Tokyo, Japan) during the test, with the results represented using the minor- and major-axis indices, as shown in Figure 2. The width was measured for 20 samples under each condition, and the results are shown with calculated error bars. The existence of race-tracking was determined by visual observation of the flow front advancement.

**Table 1.** 1 Properties of fiber reinforcement.

| Item | Properties |
|---|---|
| Tow width (mm) | 1.5 |
| Tow height (mm) | 0.5 |
| Tow volume fraction (%) | 0.64 |

**Table 2.** 1 Properties of Silicone oil used as the working fluid.

| Model No. | Specific Gravity 25 °C | Viscosity 25 °C (Pa·s) | Surface Tension (mN/m) |
|---|---|---|---|
| KF-96-100 cs | 0.970 | 0.097 | 20.9 |
| KF-96-350 cs | 0.970 | 0.340 | 21.1 |

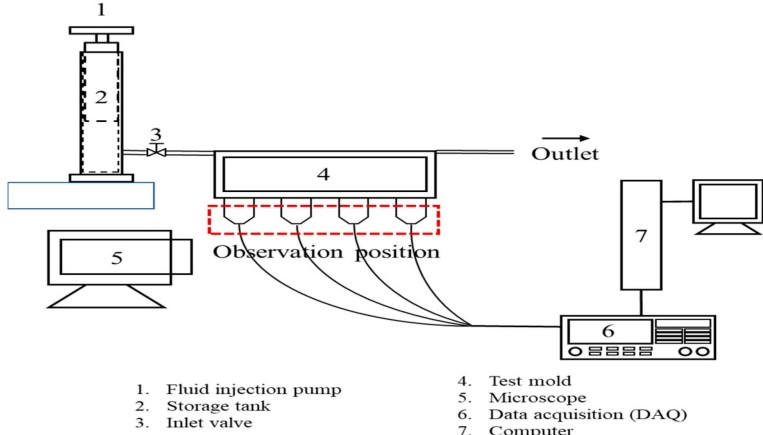

1. Fluid injection pump
2. Storage tank
3. Inlet valve

4. Test mold
5. Microscope
6. Data acquisition (DAQ)
7. Computer

**Figure 1.** A schematic of the experimental apparatus for the observation of tow behavior in double-scale porous media.

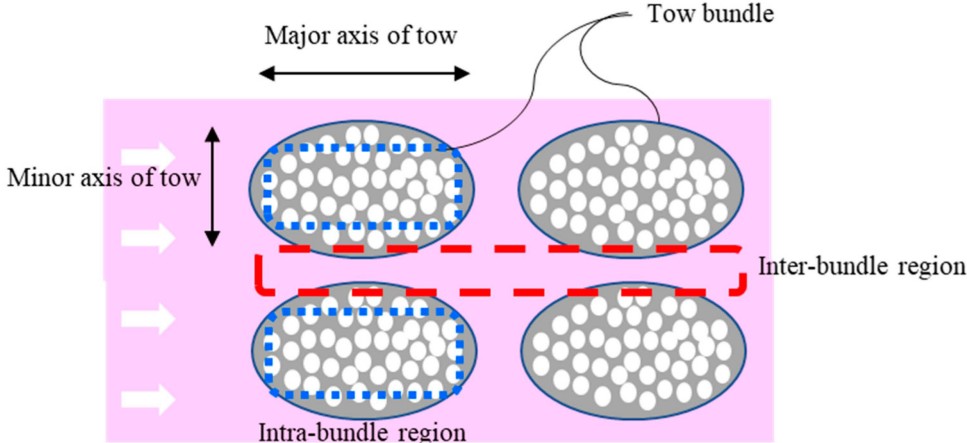

**Figure 2.** Two representative indices: tow thickness (minor axis)/the major axis, and tow bundle regions.

## 4. Results and Discussion

### 4.1. Permeability Behavior for the Porous Media under Saturated Condition

To observe the hydrodynamic effects on the reinforcement, the permeability behaviors in double-scale porous media were examined under different flow rates and viscosities.

The permeability is closely related to the pressure induced by the fluid. The hydrodynamic force induced by fluid flow is dependent on the fluid viscosity and flow rate. Therefore, the permeability behavior of the double-scale porous media was observed with fluids of different viscosities and different flow rate conditions. Figure 3 shows the permeability behaviors of the fluid in double-scale porous media with silicone fluid with viscosities of 97, and 340 cP (centipoise). The data points in increasing and decreasing condition were shown to be overlapped with some ranges. It means that permeability value has wide range of value; therefore, it is important to examine the overall tendencies under each condition. As the viscosity increased from 97 cP to 340 cP, the permeability increased by 9% on average, indicating different permeability values depending on the viscosity. The permeability also increased by 3 and 7% with increasing flow rate from 50 to 800 mm$^3$/s for the 97 cP and 340 cP of viscosity case, respectively. In the case of decreasing flow rate for the viscosity of 97 cP, the permeability decreased by 4% as the flow rate decreased from 800 to 50 mm$^3$/s with a lower value than increasing case. However, for when using silicone fluid of 340 cP, the permeability decreased by 8% as the flow rate decreased from 800 to 50 mm$^3$/s with a lower value than increasing case. Two cases showed permeability hysteresis phenomena between the increasing and decreasing flow rates of fluid in the

double-scale porous media. This may have resulted from the effect of structural network changes due to altered flow passages from the hydrodynamic force of the fluid [29].

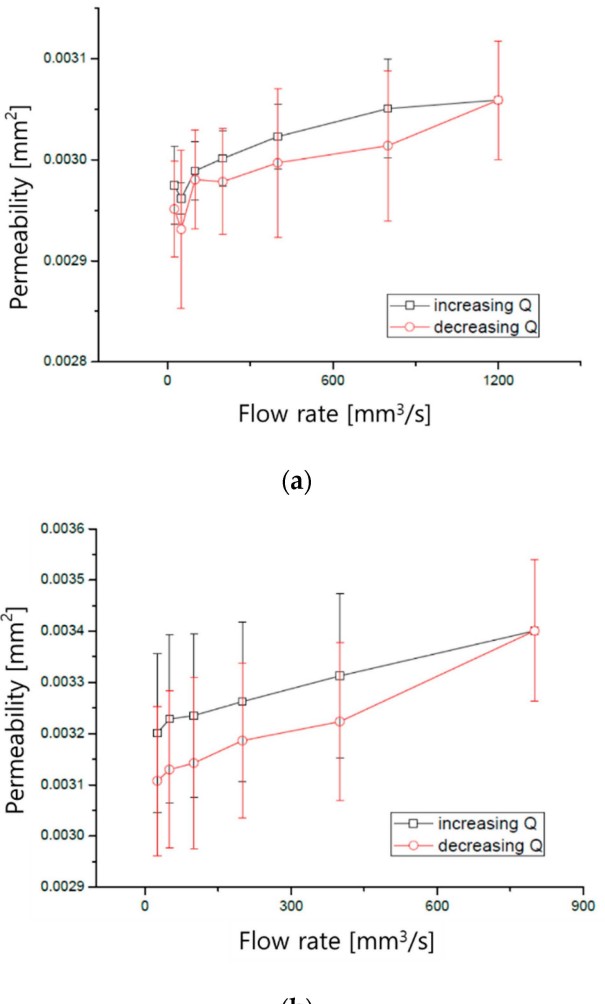

**Figure 3.** Permeability behavior with different flow rate under different viscosities: (**a**) 97 cP, (**b**) 340 cP.

*4.2. Observation of Tow Behaviors with Different Flow Rates*

4.2.1. Tow Thickness (Minor Axis of Tow)

The fibers in glass fiber-reinforced polymers can be influenced by the hydrodynamic forces induced by the fluid flow of resin during LCM. It was found that the hydrodynamic force was related to the flow rate. For the double-scale porous media, each tow consists of many single fiber filaments, as shown in Figure 4. The fiber behavior under the hydrodynamic fluid force affects the tow with the fiber filaments. Therefore, the tow behaviors of the double-scale porous media with respect to the flow rate were investigated. To examine the structural changes in the porous media, the tow behaviors were observed under different flow rates with 97 cP (0.097 Pa·s) of viscosity using cross-sections of the deformed tow and the two representative indices of the minor and major axes, noted previously, as shown in Figure 2. These two indices as functions of the hydrodynamic pressure were measured at the different positions from inlet, with an average of 20 points. In the present study, fiber bundle was assumed to no shearing.

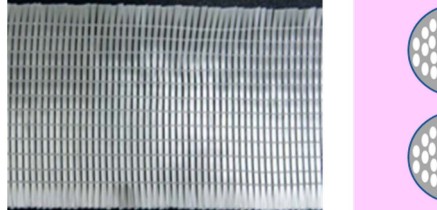
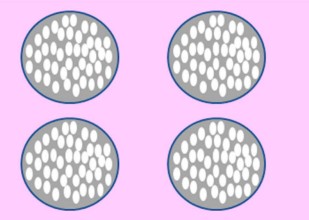

**Figure 4.** Tow illustrations with many single fiber filaments for the double-scale porous media.

Tow thickness, or the minor axis of the tow, was observed for the flow rates of 50 and 200 mm$^3$/s at the different positions from inlet, as shown in Figure 5. The most important observation was that the tow thickness decreased by 2–5% on average with increasing flow rate. In addition, the thickness of the tow decreased along the flow position. Generally, larger tow deformations are expected with increasing flow rates. However, the change in tow thickness was not prominent near the inlet. The main reason for this is the hydrodynamic entry length, which is determined by the flow rate [30–32]. At different flow rates, flows in different amounts of fully developed regions could be observed. For this reason, at a higher flow rate of 200 mm$^3$/s, the most decreased tow shape was observed at a relatively large distance of 300 mm from the inlet, compared with the distance of 180 mm for the flow rate of 50 mm$^3$/s. Furthermore, the thickness decreased more with higher flow rates because of the different hydrodynamic forces exerted by the fluid flow, as shown by the shapes of the compressed tows. Compressed relaxation behavior, in which the compressed tows returned to their original shapes, was observed as the hydrodynamic fluid force decreased. The further away from the inlet, the smaller the local fluid pressure, therefore the smaller the effect on the fiber bundle shape with fluid flow.

### 4.2.2. Major Axis of Tow

Tow deformation behavior was also observed regarding the major axis of the tow. The major axis of the tow for the flow rates of 50 and 200 mm$^3$/s was observed at the different positions, and the results are shown in Figure 6. The length of the major axis increased by 2% on average with a higher flow rate because of the different hydrodynamic forces exerted by the fluid flow. However, the greatest increase in the shape of the major axis was observed with different positions. For the same reason as the tow thickness change behavior, the hydrodynamic entry length, influenced by the flow rate, affected the tow behavior along its major axis. Compressed relaxation behavior was clearly observed at a high flow rate; these tendencies were identical to the tow thickness behavior.

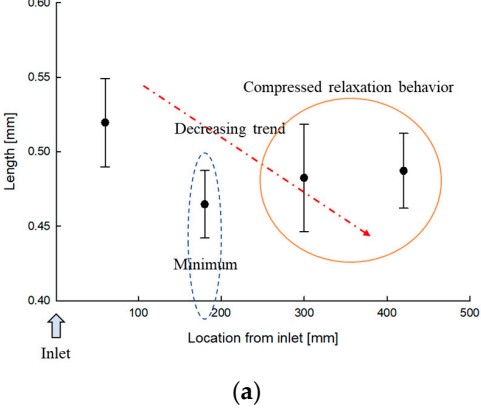

(**a**)

**Figure 5.** *Cont.*

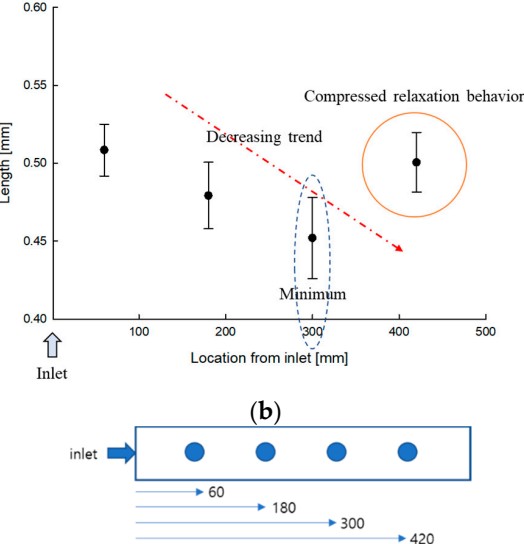

**Figure 5.** Tow thickness (minor axis of the tow) for the flow rates of 50 mm$^3$/s (**a**) and 200 mm$^3$/s (**b**) at the different positions (all dimensions are in mm).

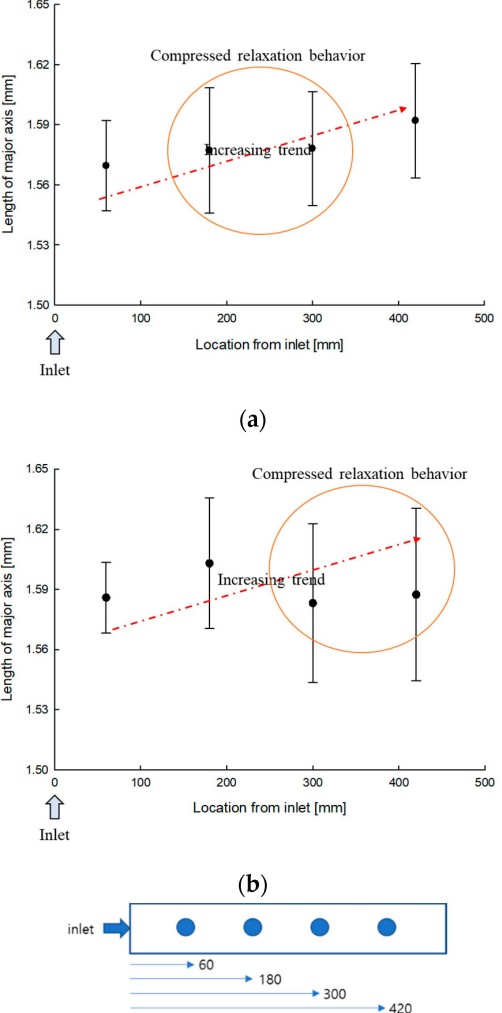

**Figure 6.** Major axis of the tow for the flow rates of 50 mm$^3$/s (**a**) and 200 mm$^3$/s (**b**) at the different positions (all dimensions are in mm).

It can be concluded that the minor axis tended to show a decreasing trend, where it showed its minimum values at different positions because of different hydrodynamic entry lengths. The major axis showed an increasing trend, with the phase showing a decreasing trend of the minor axis at the same time. This tendency was more prominent in the high-flow-rate case. The different hydrodynamic forces exerted by the different flow rates resulted in the deformed shape of the tow as described by the major and minor axes.

### 4.3. Microscopic Observation

Microscopic examination of the deformed tow behavior could provide important information on the deformation of the tow at different flow rates. The deformed tow behavior was observed during the test using an optical microscope. Microscopic observation of the cross-section of the fiber bundles during fluid flow was conducted in the O-ring sealed cavity between upper transparent glass plate and bottom mold, where non-crimped unidirectional glass fiber bundle was placed in the O-ring sealed cavity.

The length results was shown with calculated error bars which were measured at least 20 samples under each condition. Figures 7 and 8 shows representative microscopic images of the tow in the fiber bundle from the inlet to the outlet position at different flow rates, depicting transverse sections through the continuous glass reinforcement fibers. Deformed tows are seen in the fiber bundles, where both tow fibers and fibers between the tows exist, showing intra-tow and inter-tow regions, respectively. As seen in Figure 9, the flow in the double-scale porous media occurred through the reinforcement, where most fluid passed through the inter-tow regions and the remainder passed through the intra-tow region. Therefore, flow tended to be dominant in the intra-tow region, and the tow bundles became more elliptical as the flow rate increased.

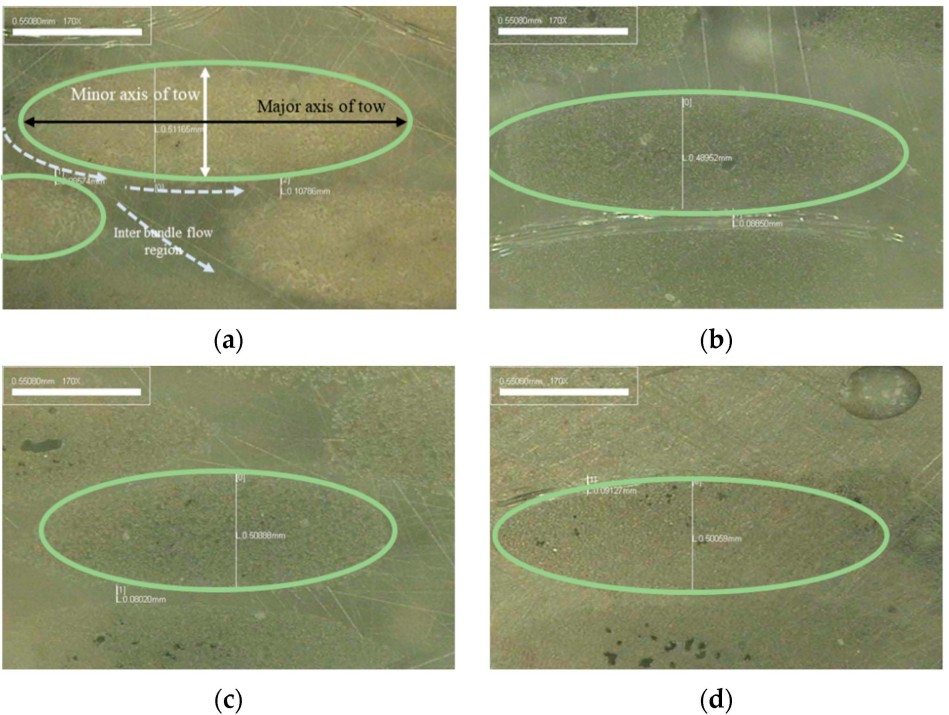

**Figure 7.** Microscopic images of the tow in the fiber bundle from the inlet to the outlet position at 50 mm$^3$/s of flow rate. (**a**): 60 mm, (**b**): 180 mm, (**c**): 300 mm, (**d**): 420 mm.

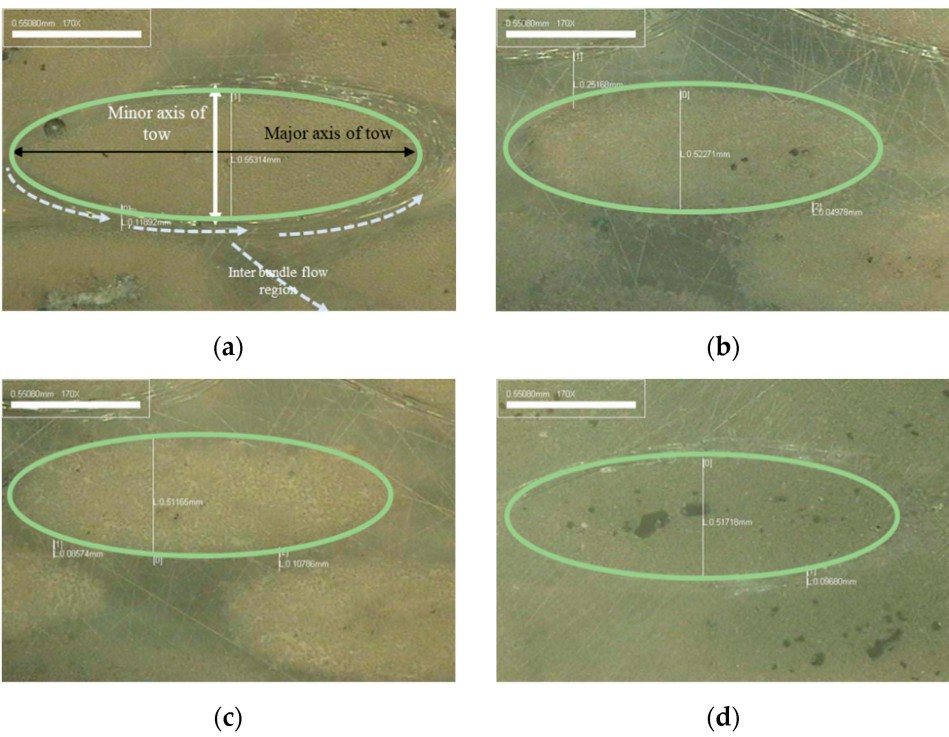

**Figure 8.** Microscopic images of the tow in the fiber bundle from the inlet to the outlet position at 200 mm$^3$/s of flow rate. (**a**): 60 mm, (**b**): 180 mm, (**c**): 300 mm, (**d**): 420 mm.

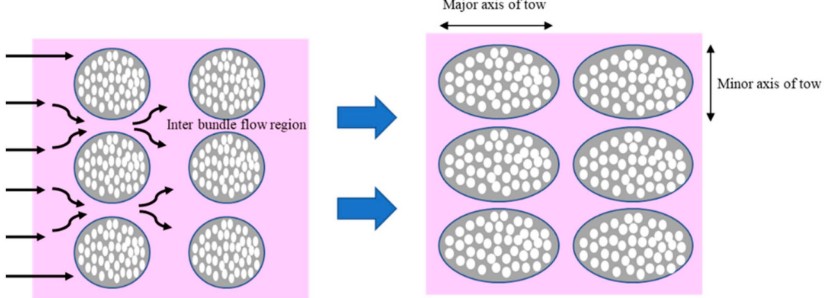

**Figure 9.** Flow behavior in the double-scale porous media.

## 5. Conclusions

In this study, tow deformation in fiber reinforcements was experimentally examined under different flow rates and viscosity conditions. The permeability behaviors in double-scale porous media were observed under different hydrodynamic conditions to examine the overall tendencies of structural changes in the reinforcing fibers. Structural change was investigated in terms of the two representative indices of tow thickness (minor axis) and the major axis. The deformed tow behavior was examined microscopically to observe the behaviors of the tow under different flow rates. The main conclusions of this study are as follows:

- Permeability increased with increasing flow rate; different permeability values were measured depending on the viscosity with increasing flow rates. Different permeability behaviors of the fluid in double-scale porous media were observed with different fluid viscosities, showing hysteresis between increasing and decreasing flow rates.
- The minor and major axes showed decreasing minimum values and increasing maximum values at different positions, respectively, which were caused by the different hydrodynamic entry lengths.

- The different hydrodynamic forces exerted by the different flow rates resulted in the deformation of the tow as described by the major and minor axis changes.
- Microscopic analysis revealed deformed tows in the fiber bundles; tow fibers and fibers between the tows showed intra-tow and inter-tow regions, respectively.
- The flow in the double-scale porous media consisted of flow passage in the tow bundles, where most fluid passed through the inter-tow spaces and the rest through the intra-tow regions. Flow tended to be dominant in the intra-tow regions, and the tow bundles became more elliptical in shape as the flow rate increased.
- The present research regarding tow deformation in reinforcing fibers can be effectively used in future research on a wide range of LCM manufacturing processes. Our future research will focus on the tow deformation in reinforcing fibers with different treatments.

**Author Contributions:** Conceptualization, S.-H.K. and S.-W.C.; methodology, M.-X.L.; investigation, S.-W.C. and S.-H.K.; resources, M.-X.L.; data curation, J.-H.Y.; writing—original draft preparation, S.-W.C. and H.-M.Y.; writing—review and editing, S.-W.C. and H.-M.Y.; visualization, M.-X.L.; supervision, S.-W.C., M.-X.L., and H.-M.Y.; All authors have read and agreed to the published version of the manuscript.

**Funding:** This research was supported in part by "Regional Innovation Strategy (RIS)" through the National Research Foundation of Korea (NRF) funded by the Ministry of Education (MOE), and in part by the National Research Foundation of Korea (NRF) grant funded by the Korean government (MSIP: Ministry of Science, ICT and Future Planning) (Grant No. NRF-2018R1C1B5086170).

**Institutional Review Board Statement:** The study was conducted according to the guidelines of the Declaration of Applied Sciences, and approved by the Institutional Review Board of Gyeongsang National University.

**Informed Consent Statement:** Not applicable.

**Data Availability Statement:** Not applicable.

**Conflicts of Interest:** The authors declare that they have no conflict of interest.

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
