# Peer review of "Tow Deformation Behaviors in Resin-Impregnated Glass Fibers under Different Flow Rates"

_applsci, doi:10.3390/app11083575_

Round 1

Reviewer 1 Report

This manuscript discusses issues related to manufacturing processes for components from fibre-reinforced polymers (composites), specifically, Liquid Composite Moulding processes. In this type of process, a liquid polymer resin is injected into a dry fibrous reinforcement.  

The authors address the deformations of fibre bundles in the reinforcements, which may be induced by the flowing polymer. This is an interesting problem, as reinforcement deformation may affect the flow of resin during the process.

However, there are several issues with this manuscript.

In general, the manuscript is not well written.

In the Introduction, a literature overview is given, which seems quite random and incoherent. In one sentence, shear tests on cross-ply laminates are referenced, in the next sentence, the simulation of multiphase flow, and then distortion of moulded composite components. It is not at all clear how all this fits together.

In Section 3, two tables are referenced. These tables seem to be missing in the document.

It is indicated that a microscope is used to determine fibre bundle cross-sections. It is not clear how this was done? During a test? After a test was finished? The information missing here would be important to understand the results.

Figures 5 and 6 seem to be identical(?)

Referring only to Figure 5, it is very hard to see any trend (unlike stated in the text).

Figures 7 and 8 seem very unconvincing. How do the authors know that the variations are not the result of stochastic variations in the fibre bundle cross-section?

Papers addressing the hydrodynamic deformation of reinforcements in LCM processing can be found in the literature. While these studies may describe reinforcement deformation at a larger scale, they are still related to the problem discussed here and should have been cited.

In summary, the manuscript is not suitable for publication.

Additional comments:

Abstract

It is not clear what the “rapid development of high-performance fibres” refers to. Fibres typically used in composite components, e.g. carbon fibres, have been around for decades(?)

Section 2

Equations (2) and (3) relate to uni-directional injection experiments. Other equations apply to different flow geometries.

Author Response

This paper has been reconstructed as the reviewers’ suggestion. We, the authors, really appreciate the reviewers’ helpful comments. We feel that this paper has become more rigorous and well-constructed after implementing the suggested parts in the manuscript. The modified or added parts are highlighted in yellow background in the manuscript.

Reviewer 2 Report

Review Report is attached.

Author Response

(The authors gave the same response as above.)

Round 2

Reviewer 1 Report

This is a revised version of a manuscript discussing deformations of fibre bundles in reinforcements, which may be induced by flowing polymer in LCM processes.

The authors made some revisions, but the text still lacks clarity. Important information seems to be missing. The results seem not very convincing (maybe because they are not explained well).

The manuscript does not warrant publication.

Some detailed comments:

In general, the manuscript is still not well written.

Page 2, line 14

Is the cycle time in LCM processes not short compared to other processes, e.g. autoclave processing?

Page 2, lines 30-40

There are several publications addressing specifically the problem of hydrodynamically induces deformation of the reinforcement (e.g. Hautefeuille et al., Bodaghi et al., Endruweit et al.).

Page 2, line 51 etc

What is discussed here is the deformation of fibre reinforcement through shear (during preforming). This effect is not relevant here.

Page 3, line 28

Q and A do not appear in Eq. (1).

Page 4, line 11

Generally, capillary effects are thought to be the main reason for differences between saturated and unsaturated permeability.

Page 4, line 20

In the text, the horizontal dimension of the fibre bundles is given as 1.5 mm. In the table, the value is 1.6 mm. Why is it different?

Page 4, line 35

Should it read “width” rather than “length”?

Table 1

Should it read “tow width” in the first row?

Table1

What is the difference between “tow volume fraction” and “fibre volume fraction of tow”?

Figure 3

Considering the error bars, are differences between the data points (statistically) significant?

Figure 5

It is still not clear what the figure shows. Where was the flow front? And at what position was the bundle cross-section measured? Was it at the flow front? Or behind the flow front?

Certainly, the pressure at the flow front is always the same?

If the bundle dimensions were measured behind the flow front: The further away from the injection gate, the smaller the local fluid pressure, i.e. the smaller the effect on the fibre bundle shape.

Figures 7, 8

It is still not clear how the cross-section of the fibre bundles was imaged. How was it possible to do this with the experimental set-up shown in Fig. 1?

Author Response

(The authors gave the same response as above.)

Reviewer 2 Report

Minor correction is needed:

  1. Figure 2, in line 6 of page 5: The inter-tow region is the one between the tows (bundles), while the intra-tow flow region is the one between fibres (filaments). Please modify it, since you did the opposite.
  2. At Figure 7 and 8: you illustrated the fibre bundle's thickness (minor axis) changes as well as the inter-tow flow region (passages) . Please demonstrate also the fibre bundles' width (major axis) changes.

Author Response

(The authors gave the same response as above.)

Round 3

Reviewer 1 Report

This is another revised version of a manuscript discussing deformations of fibre bundles in reinforcements, which may be induced by flowing polymer in LCM processes.

The authors made revisions addressing the detailed comments on the previous version. The manuscript is acceptable for publication.